# A Monosodium Iodoacetate Osteoarthritis Lameness Model in Growing Pigs

**DOI:** 10.3390/ani9070405

**Published:** 2019-07-01

**Authors:** Joost Uilenreef, Franz Josef van der Staay, Ellen Meijer

**Affiliations:** 1Department of Clinical Sciences of Companion Animals, Faculty of Veterinary Medicine, Utrecht University, Yalelaan 108, NL-3584 CM Utrecht, The Netherlands; 2UMC Utrecht Brain Center, University Medical Center Utrecht, Utrecht University, Universiteitsweg 100, NL-3584 CG Utrecht, The Netherlands; f.j.vanderstaay@uu.nl (F.J.v.d.S.); e.meijer1@uu.nl (E.M.); 3Behavior and Welfare Group, Department of Farm Animal Health, Faculty of Veterinary Medicine, Utrecht University, Yalelaan 107, NL-3584 CL Utrecht, The Netherlands

**Keywords:** lameness, gait analysis, Monosodium Iodoacetate, osteoarthritis model, within-subject analysis, synovial tissue histology, model validation

## Abstract

**Simple Summary:**

Osteoarthritis is a cause of lameness in pigs. It causes pain for affected pigs and reduces profit for the farmer. From an acute problem, it progresses into a chronic condition. To study treatments for osteoarthritis, a model that mimics the functional and structural aspects of osteoarthritis is needed. To induce osteoarthritis, we injected a chemical compound (Monosodium Iodoacetate) into the carpal joint of 10 pigs (treatment group). Ten other pigs were injected with an innocuous substance (control group). We assessed their gait by visual inspection and with a device that measures weight-bearing on each limb. After 68 days, we euthanized the pigs and examined the tissues of injected joints microscopically. We could confirm structural joint changes resembling osteoarthritis in 8 of 10 pigs in the treatment group These pigs also placed less weight on their affected limb compared to the control group on day 3, 14, 28, and 56. Visually, these pigs were only more lame on day 1. Treatment with Monosodium Iodoacetate caused joint changes and lameness resembling those of naturally-occurring osteoarthritis. Although the model needs improvement for use in visual lameness, it enables the study of (drug) intervention on objective movement–pain behavior and structural joint changes.

**Abstract:**

Lameness is a common problem in pigs, causing welfare issues in affected pigs and economic losses for farmers. It is often caused by osteoarthrosis (OA) in its acute or chronic form. We assessed face and construct validity of a potential model for naturally-occurring OA and its progression to chronic OA. Such a model would allow the assessment of possible interventions. Monosodium-iodoacetate (MIA) or isotonic saline was deposited in the intercarpal joint of 20 growing pigs. Functional effects were assessed using subjective (visual lameness scoring) and objective (kinetic gait analysis) techniques at several timepoints. Structural effects were assessed by histopathology at 68 days. Eight out of 10 MIA treated animals had histopathological OA lesions confirmed in the target joint, while for all saline treated animals the target joint was judged to be normal. Pressure mat analysis revealed increased asymmetric weight bearing in these animals compared to the control group on day 3, 14, 28 and 56. Visual scoring only showed a difference between groups on day 1. MIA did not cause prolonged visible lameness, thus face validity for OA under field conditions was not entirely met. Since objective gait parameters showed decreased weightbearing as a behavioral expression of pain, it may be used as a general model for movement-induced pain in pigs.

## 1. Introduction

Lameness is a common problem in swine husbandry [1]. It causes welfare issues in affected pigs, such as pain and diminished competitive capabilities. Financial damages caused by lameness in livestock may also be substantial [2,3,4,5].

Osteoarthritis (OA) is a frequent cause of pain-induced lameness in pigs. The lameness is mostly due to pain, although in advanced stages extensive bony changes may also cause mechanical impairment. Like the changing views on the pathogenesis of OA in humans, OA in pigs should not be considered a single disorder but as a group of different or partially overlapping disease phenotypes that share certain clinical and pathological features [6]. OA in pigs is most often caused by infectious agents, physical injury, or disrupted skeletal development in the form of osteochondrosis [7]. Regardless of the starting point of synovial tissue insult, the reaction patterns of these synovial tissues are limited. If the initial insult or ensuing synovial tissue damage is not resolved, progression of the condition will follow along convergent lines toward end stage OA, albeit with variable timelines. As the interest in pig OA is just starting to increase there is a paucity of studies published, and the age-related OA phenotype has only recently been described [8]. 

OA presents itself as articular cartilage degeneration or damage, with a mild or excessive synovial inflammatory response, which may be accompanied with marked joint distention. Articular cartilage will start to show fissures and ulceration and eventually subchondral bone becomes exposed. The subcortical plate thickens, and osteophytes may form [8].

To minimize the consequences of OA in affected animals, the condition must be detected as early as possible [9]. This maximizes the window to slow down or, provided effective disease modifying interventions become available, halt progression towards clinical end-stage OA. 

To study the effects of interventions, either animals with naturally-occurring disease or animals in which the disease is modelled experimentally can be used. Naturally-occurring disease has the advantage that all aspects of the disease are present, including its underlying pathophysiological processes. There are however also disadvantages. The occurrence of the disease may be unpredictable, as is the location of the lesions. It may also be difficult to diagnose early phases of the condition in vivo, because of a lack of validated sensitive diagnostic methods to identify appropriate animals before overt behavioral changes are apparent. The systematic study of the effects of interventions on the progression of OA therefore warrant the use of an experimental model mimicking both the acute and chronic phase of OA. 

A single animal model is unlikely to be appropriate to study all aspects of a condition. The validity for studying certain aspects of a condition in the model being considered therefore needs to be established before it can be used. In biomedical research, the validity of a model is commonly evaluated using three distinctive validation criteria that address face, predictive, and construct validity [10].

Face validity refers to what the model should resemble on general impression or easily discernible presentation, i.e., it should be phenotypically similar to the condition it is supposed to model. However, this does not automatically imply that the underlying pathobiology is indeed the same or even comparable to that of the modelled disease state. Predictive validity of an animal model is inferred from the ability of the model to predict a specified outcome, such as behavior or histopathology, or the efficacy of a therapeutic intervention in the specific situation it is supposed to mimic; the latter being predictive validity in the narrow sense such as used in pharmacological research. These predictions would then allow extrapolation of an observed outcome to other situations. 

Construct validity refers to the legitimacy of either the experimental manipulations used or the measurements that are performed in a model to assess a certain aspect of the condition. A model with strong construct validity therefore shows a strong relationship between the best theory available regarding the etiology or pathogenesis of the target disorder [11].

The validation of a putative animal model is an iterative process in which several cycles of building, assessing and evaluating the model may be needed until the validity of a model is established [11]. As can be inferred from the previous descriptions of different aspects of validity, it is very important to decide and report on the intended use of the model and the specific aspects of the condition it is supposed to model. In this study, the goal was to develop an inducible model that in standardized laboratory settings leads to predictable pathological changes in the joint with the associated behavioral expression lameness. This would enable the assessment of interventions that minimize the welfare issues associated with OA in pigs, which are mostly due to pain and the resulting lameness. A model would therefore not only have to induce histopathological changes that resemble OA, but also produce pain-induced lameness.

Osteoarthritis is depending on the stage characterized by at least two distinct types of measurable pain. That these two types of pain are distinct types is demonstrated by the differential sensitivity to (analgesic) drug treatment [12,13]. One is non-ambulatory pain, which occurs spontaneously or due to nociceptive sensitization processes both peripherally and more centrally. With the latter, a non-noxious movement unrelated stimulus like heat or gentle pressure (termed quantitative sensory testing) may provoke the display of nocifensive reflexes or outwardly expressed conscious pain behavior. However, nocifensive reflexes may not represent pain per sé and detecting conscious pain behavior requires knowledge of species-specific pain behavior, which for pigs was recently reviewed [14].

The second distinct measurable pain type is movement induced pain, or in non-verbal subjects movement induced nociception, when outward behavioral signs of conscious pain are not readily detected. With this pain type, sensory input provoked by movement and/or dynamic weight bearing results in detectable changes in locomotor patterns [15,16] sometimes accompanied with an outwardly display of conscious pain behavior.

To quantify lameness in pigs, several methods are available, of which visual scoring is the simplest, fastest, and least expensive. This is also the method that is used in the field by farmers and veterinarians for assessing locomotion in pigs [17]. However, it is subjective [18], suffers from only moderate inter-observer agreement [19,20] and does not always correlate well with either objective kinetic and kinematic measurements or joint pathology [21,22]. Therefore, we also assessed limb loading using kinetic pressure mat analysis [23,24,25]. To induce OA, we required a model in pigs that would result in (1) an adequate timespan of pain-induced lameness, (2) discernable reproducible histopathology, and (3) in which the impact of an early intervention on the progression of the condition could be assessed. For the pig, only a mild, reversible inducible synovitis model has been described [23]. In rats, intra-articular injection with monosodium-iodoacetate (MIA) has been established as a robust model for osteoarthritic joint change and pain-associated lameness, with the desired reproducible progression and time-span [25,26,27].

Intra-articular deposited MIA interferes with cartilage metabolism by inhibition of chondrocyte glycolysis [28,29]. Over time, it induces dose-dependent changes in chondral and subchondral tissue, and with higher doses results in clinical manifestations of osteoarthritis, including decreased weight bearing [26], decreased spontaneous locomotor activity and secondary progressive long-term loss of spontaneous mobility [25]. Consequently, we expect that MIA is a potentially useful method to induce OA in a porcine model.

In this study, we therefore assessed face and construct validity of MIA-induced lameness as a model for naturally occurring lameness due to OA in pigs, using gait scores, pressure mat analysis, and histopathology. We expected that the pigs showed phenotypically a symptomatology that is comparable to that of naturally occurring OA (face validity). For the model to have construct validity, the (histopathological) changes in the animal that cause the altered phenotype (in this case, lameness) should be comparable to that in naturally-occurring OA. Therefore, we expect to see cartilage damage, inflammation of the synovial membrane and damage to subchondral bone. Also, we expected that these changes induced pain in the affected joint which in turn caused lameness. We did not intend to mimic the naturally occurring processes that induce OA.

## 2. Materials and Methods 

### 2.1. Ethical Statement

The study design, including a standard rescue analgesia protocol for severely lame animals as well as an exploratory dose-response study in 8 piglets (please, see Appendix A), was reviewed and approved by the local ethical committee of Utrecht University (no. 2014.I.11.085, date of approval 17 December 2014), The Netherlands, and was conducted in accordance with the recommendations of EU directive 86/609/EEC. All effort was taken to minimize the number of animals used and their suffering. The analgesic rescue protocol for severely lame animals consisted of 0.01 mg/kg buprenorphine IM (0.3 mg/mL, Temgesic, Reckitt Bensicker BV, Hoofddorp, The Netherlands) every 12 hours as required. 

### 2.2. Animals

At weaning at the age of 28 days, 20 piglets (Topigs 20 × Pietrain cross), born at the teaching farm of Utrecht University, were selected from a total of 5 litters (Table 1). All piglets underwent a general clinical examination and visual assessment of locomotion by a veterinarian at the time of recruitment. The assessment was based on the behavior, posture, gait, nutritional status, upkeep of coat, demeanor, and absence of prominent clinical abnormalities, as described in detail in a standard textbook [17]. Only healthy, sound piglets were selected. The selected piglets (4 per litter) were transported to the research stable of the Behavior and Welfare group.

### 2.3. Housing

Piglets were group-housed with littermates in pens with solid concrete floors and deep straw bedding. Each pen contained an adjustable covered nest area, fitted with rubber flaps to provide extra shelter during the first weeks. Each nest area had one heat lamp per 2 m^2^ and the floor was covered with a rubber mat with abundant straw bedding on top. Enrichment, consisting of metal chains, balls, and rubber chewing sticks, was provided for the entire duration of the study. Water was available ad libitum through a drinking nipple. The pigs were fed commercial standard food for growing pigs (Stimulans, De Heus, The Netherlands) ad libitum in a large food trough. A radio was playing softly in the stable day and night, including during testing.

### 2.4. Handling and Habituation

All husbandry related activities and general welfare checks were performed daily by qualified personnel that the pigs were familiar with. Additionally, personnel spent at least one hour sitting quietly in the pen, touching and petting pigs that approached voluntarily, until all pigs could be touched without evoking avoidance reactions by the pigs. Piglets were habituated to the experimental set-up with their entire group first. When they did not show any outward signs of distress such as vocalizations or escape attempts the groups were made progressively smaller until animals were able be alone in the experimental set-up 3 weeks later. To minimize the time needed to collect successful trots for the videotaped open space locomotion as well as the for the pressure mat runway, the pigs were trained before the start of the experiment using positive reinforcement with clickers and sugar-coated milk chocolate treats (M&M’s^®^, Mars Chocolate, Veghel, Netherlands) as rewards. Normal exploratory behavior across the runway was shaped until each pig was able to trot down the runway in a straight line at constant speed without stopping. Training sessions for piglets isolated from their pen mates lasted for a maximum of 10 minutes per pig per session.

### 2.5. Experimental Design

Piglets from each litter were randomly assigned at day 0 (envelope method) to either the MIA treatment (MIA-group; n = 2 per litter, total over 5 litters: n = 10) or the isotonic saline control group (SC-group; n = 2 per litter, total over 5 litters: n = 10). All personnel, with the exception the researcher performing the intra-articular injections (JU) remained blinded for treatment. 

Pressure mat measurements as well as videos of the locomotion of the pigs were collected on several timepoints (Table 2). Visual scoring was performed off-line from the videos to keep the observers blinded with respect to the timepoint and group.

#### 2.5.1. Intra-Articular Injection

Intra-articular injection technique was practiced with methylene blue on post-mortem material of pigs of the same size as the experimental animals (Appendix A). The dose of MIA that was injected into the intercarpal joint (20 mg as an 80 mg/mL solution) was based on an in vivo pilot study using three doses: 10 mg, 20 mg and 40 mg (Appendix A).

Anaesthesia, aseptic intra-articular injections and recovery were all carried out in dedicated rooms separated from the housing pens. At day 0, animals were guided from their housing pen into a dedicated pen for induction of anaesthesia. Animals were sedated 2 at a time with dexmedetomidine IM (15 µg/kg, Dexdomitor 0.5 mg/mL, Orion Pharma, Espoo, Finland). Following clinical sedation, general anaesthesia was induced using ketamine (10 mg/kg, Narketan 10, Vétoquinol S.A., Magny Vernois, France) and midazolam (IM; 0.5 mg/kg, Midazolam Actavis, 5 mg/mL, Actavis Group PTC, Hafnarfjördur, Iceland) in the same syringe. Supplemental flow by oxygen (8 L O_2_/min) was provided during the procedure and in recovery. Under general anaesthesia, the animal was transported to the procedure room. The left carpal joint was aseptically prepared by clipping, followed by a 2-step removal of hair and debris using an iodine-based scrub solution, and disinfection with 70% alcohol and chlorhexidine (Hibisol, Regent Medical Ltd., Nottingham, UK). Pigs received a slow intra-articular injection with either 0.25 mL MIA (80 mg/mL solution, sodium iodoacetate BioUltra 98%, Sigma Aldrich, St. Louis, MO, USA) or 0.25 mL of sterile NaCl 0.9% (B. Braun Melsungen AG, Melsungen, Germany). The left carpus was slightly bended in a 100° to 110° angle, under gentle distention, and the needle was inserted dorsally, just medial from the central axis, 2 to 3 mm deep, into the proximal intercarpal joint. After deposition of the injectate, the carpus was slowly flexed and distended, before animals received atipamezole (IM; 0.3 mg/kg Atipam 5 mg/mL, Eurovet Animal Health, Bladel, Netherlands) and were transported to one of 2 adjacent pens dedicated for recovery. Soft flooring and external warmth were provided until animals were able to walk back to their housing pen. 

#### 2.5.2. Locomotion and Gait Measurements

##### Visual Scoring

Two experienced observers, one board-certified porcine health specialist/ECPHM (European College of Porcine Health Management) diplomate and one ECPHM resident, visually scored all pigs for lameness. They were blinded for both treatment and experimental day by providing them with off-line 2–3 min movie clips of each pig at baseline, day 1, 3 and 28, striped of ID and timepoint. Animals were encouraged to stand, walk and trot over a 10 to 15 m distance and were filmed from all sides. Observers could watch the movies as often as they needed to reach a decision on the score, but they could not use slow motion or freeze the movie at any time. Visual scoring was performed according to a protocol modified from Main et al. [30] (see Table 3). To maintain blinding for pig ID however, the protocol was adapted in which all descriptors that incorporated behavior (requiring pig ID within the home pen) were removed. 

##### Pressure Mat Analysis

A Footscan^®^ 3D Gait Scientific 2 m pressure mat system (RSscan International, Olen, Belgium) with an active sensor surface of 1.95 × 0.32 m containing 16,384 sensors (2.6 sensors per cm^2^), with a sensitivity of 0.27–127 N/cm^2^ and a measuring frequency of 126 Hz was connected to a laptop with dedicated software (Footscan Scientific Gait 7 gait 2nd generation, RSscan International, Olen, Belgium). Calibration according to the manufacturer’s specifications using a person weighing 62 kg preceded each measuring session. The mat was mounted flush with a 40 × 1000 cm runway [31]. A holding pen with a guillotine door was located at both ends of the runway (Figure 1). The entire runway was covered with a rubber mat (5 mm thick, shore value 65° ± 5). All runs were videotaped using 2 action cams (JVC-GC-XA1, JVC, Kawagana-ku, Yokohama, Japan) on both ends of the runway. The cameras were mounted 2 meters above the runway and each faced down to the middle of the runway, while capturing the entire runway ‘in frame’. A third camera (Panasonic HDC-SD9, Matsushita Electric Industrial Co. Ltd, Kadoma, Japan) was used to videotape the lateral view of the trot for visual scoring.

On measuring days, pigs from one pen were placed in a holding area adjacent to the runway. The pig in the runway was able to hear and smell its penmates but could not see them. Pigs were let into the runway one by one and tested in the order they presented themselves in. Two people at either side of the runway operated the guillotine doors and used clickers and chocolate treats to reward the pigs. A third investigator, blinded to treatment, operated the software and judged each run on the following criteria: Pigs had to trot the entire length of the runway at a visually steady pace in either direction in a straight line and looking straight ahead. Runs that fulfilled these criteria were saved and the process was repeated until 4 runs meeting these criteria were collected. All pigs performed the required 4 valid runs within 5 min.

#### 2.5.3. Euthanasia

General anaesthesia of the animal was induced under identical conditions as described preceding the intra-articular injections. When clinical depth of anaesthesia was judged as adequate, IV access (ear vein, jugular or superficial abdominal vein) was secured using an IV cannula and the pigs were euthanized by IV injection of 50 mL of Pentobarbitone (400 mg/mL, Euthanimal, Alfasan, Woerden, The Netherlands). After euthanasia, the pigs were transported to the Department of Pathobiology of the Faculty of Veterinary Medicine of Utrecht University.

#### 2.5.4. Necropsy and Histopathology

Within hours of euthanasia, a full necropsy including opening of left and right carpal joints, shoulder joint, elbow joints, knees, and tarsal joints was performed. Samples of ca. 4 mm thick were taken of both radiocarpal joints using a K430 band saw (Kolbe GmbH FOODTEC, Elchingen, Germany) equipped with 16 × 0.4-4TPI blades (AB Munkforssåger, Munkfors, Sweden). These samples were placed in 10% neutral buffered formalin and stored at room temperature until sufficiently fixated. After fixation the samples of the joints were decalcified in 10% ethylene-diamine-tetra-acetic acid (EDTA, Sigma Aldrich, St. Louis, MO, USA). Decalcification time varied between 2 and 15 weeks. Hereafter the samples were embedded in paraffin, cut into sections of 3 micrometer, stained with haematoxylin and eosin (HE) and evaluated by light microscopy (Olympus BX-45, Zoeterwoude, The Netherlands) by a board-certified veterinary pathologist.

### 2.6. Data Preparation

Claw strikes were manually assigned to left front (LF), right front (RF), left hind (LH), and right hind (RH) limbs using the Footscan software. Peak vertical force (PVF (N)) and Vertical Impulse (VI (Ns)) were calculated from the raw data by the program. A left-right asymmetry index (ASI) comparing the two front limbs was calculated using the following equations:(1)$$ASI_{VI}\;=\;\frac{VI_{LF}\;-\;VI_{RF}}{0.5\;\times \;(\;VI_{LF}\;+\;VI_{RF})}$$
(2)$$ASI_{PVF}=\;\frac{PVF_{LF}-PVF_{RF}}{0.5\;\times \;\left(\;PVF_{LF}+PVF_{RF}\right)}$$

An ASI of 0 indicated perfect symmetry and the extreme values of −200 or +200 indicated non-weight-bearing lameness on the left and right side, respectively.

### 2.7. Statistical Analysis

A linear mixed effects model was used to evaluate the effect of day, treatment, litter, and gender and their interactions as fixed factors on ASI of PVF and VI. Pig was treated as random effect. Model selection was based on Akaike’s information Criterion (AIC), and the appropriateness of the models was evaluated by a visual inspection of graphs and residues for normality and homoscedasticity. 

Agreement between two observers assigning visual scores was assessed using Cohen’s weighted ƙ with squared weights of differences.

The results of visual scoring were compared between groups for each timepoint using Fishers’ exact test. 

All analyses were performed using R Statistical software version 3.1.2 (R. Foundation of statistical Computing, Vienna, Austria) [32] with package nlme [33].

Statistical significance was set at *p* < 0.05. Unless indicated otherwise, results are presented as mean +/− SD.

## 3. Results

All animals were judged to be clinically healthy and of sound locomotion at the time of recruitment up to the point of the baseline Footscan measurement.

Marked clinical lameness (lameness score 4 or higher) was observed 4–5 h after intra-articular injection in 8 animals, upon which they received as per protocol rescue analgesia consisting of 0.01 mg/kg of IM buprenorphine (0.3 mg/mL, Temgesic, Reckitt Bensicker BV, Hoofddorp, Netherlands) None of the animals met the criteria for subsequent doses of rescue analgesia during the study. After unblinding, these 8 animals were found to be in the MIA-group. 

### 3.1. Pathology

#### 3.1.1. Gross Pathology

All 20 animals were found to be in excellent conditions and no visual signs of internal organ changes were observed. In the control group, in 4 animals a singular slight depression of articular cartilage was observed macroscopically in the opened left stifle (n = 3) and right shoulder (n = 1). 

None of the 20 target joints showed an increase in synovial fluid. In the MIA treated group, 1 animal did not show osteoarthritic changes in the target joint. In the remaining 9 animals, macroscopically the left radiocarpal joint surface was severely irregular and dull (cartilage necrosis), with multifocal red discolorations (Appendix A, panel b). In the 9 damaged target joints, synovial membrane thickening (n = 3), thickened joint capsule (n = 1) and subchondral bone necrosis (n = 4) were observed macroscopically, but all in different affected target joints. Macroscopically, none of the animals in the MIA groups showed changes in other opened joints.

#### 3.1.2. Histopathology of Target Joints

In 5 control animals, either in the left (n = 2) or both (n = 3) carpi focal changes were detected within the articular cartilage, characterized by solitary necrotic chondrocytes, shrunken and hypereosinophilic chondrocytes and loss of chondrocytes, compatible with spontaneous osteochondrosis latens. In the MIA treated group, 1 animal did not show histopathological lesions. This was the same animal that also did not show macroscopic lesions. From another animal, showing macroscopic lesions, the sample for histopathological evaluation could not be recovered, due to technical difficulties during a much prolonged decalcification and preparation phase. In the remaining 8 macroscopically damaged carpi of which a sample for histopathology was available, the joint surfaces were moderately irregular. The articular cartilage lost its hyperchromatic basophilic staining, and multifocally chondrocytes were either shrunken and hypereosinophilic, or not present at all (necrosis). Occasionally there was formation of fissures and cystic spaces in the necrotic cartilage. These necrotic areas were multifocally surrounded by vital cartilage in which clusters of hyperplastic chondrocytes (chondrones) were visible. Multifocally the subchondral bone was replaced by hypereosinophilic amorphic material in which shrunken, hypereosinophilic, necrotic osteocytes were present. Within the bone marrow surrounding these areas mildly increased amounts of osteoclast and moderately increased amounts of osteoblast were seen. Multifocally both cartilage and subchondral bone were replaced by fibrovascular tissue.

In summary, changes in these joints were characterized by chondral erosions, chondrocyte necrosis, the formation of chondrones and exposure of subchondral bone.

### 3.2. Gait Analyses

#### 3.2.1. Visual Scoring

Due to technical difficulties, one movie was not available for scoring lameness. This was a movie of a pig from the MIA group on day 3. 

There was substantial agreement between the two observers, ƙ = 0.614, *p* < 0.005.

Lameness scores 1 and 2 were attributed to pigs in both the MIA and SC groups and on all days. Lameness score 3 and 4 were only given to pigs in the MIA group (Figure 2). Fishers exact test showed a difference in visual scores between groups only on day 1, with visual scores being higher in the MIA group (0.81+/−0.54) compared to the SC group (0.35+/−0.48). The pig from the MIA group that did not show any (histo-)pathological lesions had been assigned a gait score of 0 for all timepoints by both observers. The pig with macroscopic lesions but a histological sample being unavailable due to technical difficulties, had been assigned gait score 1 at day 3 and 28 by 1 observer, and score 0 for all other timepoints, including those assigned by the other observer.

#### 3.2.2. Pressure Mat Analyses

Data from two animals from the MIA group were excluded from analysis, on account of no histopathological joint changes (n = 1) and no histopathological sample available (n = 1) to verify osteoarthritic changes observed on macroscopy. Pre-injection, ASI of VI and PVF in both groups were close to 0 (0 indicating perfect symmetry). Apart from ASI of PVF on day 7, when ASI scores were similar (and in the control group, close to 0), PVF and VI values were lower on all days post-injection (i.e., the left front limb was loaded less than the right front limb). In addition, the MIA group had lower ASI’s of PVF and VI than the SC group on day 3, 14, 28, and 56 (Table 4, Table 5 and Table 6).

## 4. Discussion

Choosing an appropriate in vivo animal model to study OA depends on which aspects of OA development are targeted and what the target species are [8]. We selected the pig as model species for studying pain related lameness under experimentally controlled conditions, with a clear veterinary focus. However, an OA model in the pig might also be useful to study the behavior of peri-articular structures in human OA, as proportions and intra-articular cartilage and associated ligaments are considered to have characteristics close to the analogous structures in humans [34]. Furthermore, pigs have a digestive tract more similar to humans than most other species used in in vivo animal OA research. Translational aspects from pig to human might therefor be more successful in the case of orally administered medication as interventional therapy [34].

In our study, we injected MIA to induce osteoarthritis in a single joint of pigs as a controlled experimental intervention. We tracked functional characteristics relative to baseline and a saline- control group using both visual scoring and limb loading kinetics up to day 56 after injection, with subsequent full necropsy and histopathology one week following completion of data collection. It has previously been shown in pigs that decreased weight-bearing is a useful measure in both experimentally-induced [23,24] and naturally-occurring [15,31] lameness. To our knowledge, the effects of MIA on weight-bearing in pigs have not been described before.

As between species translational aspects of OA are not the focus of the present study, the interested reader is referred elsewhere [35,36].

Next to face and construct validity of our model, the longitudinal within subject design of the present study also merits a limited discussion of predictive validity for naturally occurring lameness in pigs.

### 4.1. Face Validity

Face validity refers to how well the model resembles aspects of the disease condition phenotypically. We focused on histopathological joint alterations as the main histological endpoint, and altered weight bearing as the main clinical endpoint.

#### 4.1.1. Histopathological Presentation of Joint Changes

Eight out of ten animals which were administered MIA showed histopathological changes in the injected joint, corroborating the MIA-induced synovial tissue changes described in other species [25,27,37]. MIA injection failed to induce histopathological changes consistent with OA in one animal and in another animal a sample for histopathological evaluation was lost due to technical difficulties. In the present study, histopathology was performed at a single timepoint after intra-articular injection, therefore dynamic changes in the injected joint caused by MIA could not be assessed. 

When induction of MIA-associated histopathological OA joint changes is depending on a single MIA injection, the total dose of MIA administered [38] and intensity of exercise regime [39,40] have been shown to severely limit the extent or even completely reverse typical histopathological joint changes associated with MIA observed on necropsy. In our present study, the pigs received more exercise than an average pig on a commercial farm. Pressure mat measurements took on average 5–10 min per pig, although most of this time was spent consuming the rewards and waiting until the measuring equipment was ready for another run. Additionally, the pigs were housed under enriched conditions with straw bedding and more space than commercially housed pigs, which may have resulted in more activity and thus more exercise. 

Although the effect of an exercise regime in our prospected pig model warrants further study, we argue that the most likely explanation for the absence of typical MIA -associated histopathology in two animals is failure to introduce or contain a sufficient dose of intra-articular MIA. This despite diligent practice of the intra-articular injection technique on same size piglet cadavers before applying this technique in the animals in our study. 

#### 4.1.2. Altered Weight Bearing (Lameness Assessment)

We assessed lameness by two methods: Subjective evaluation by veterinarians using a visual scoring system [30], and objective quantification of limb loading using a pressure mat [31]. Visual scoring is the only method routinely used in commercially housed pigs, and therefore relevant to daily practice. Until more sensitive diagnostic methods become widely available, pigs that are visually clearly lame will be identified by farmers and veterinarians and only this subpopulation with pronounced clinical lameness will be considered for treatment. Subclinical stages of OA are not detectable using this method. In order to also evaluate objective parameters for lameness in our model, pressure mat analysis was performed, a method that was expected to be more sensitive than the visual scoring method [41].

##### Visual Scoring

The visual scoring data only detected lameness between in the MIA-group compared to SC-group on day 1. The agreement between the two blinded observers using the adapted scoring-system was ƙ = 0.614. In the publication on the original scoring system, agreement varied between virtually no agreement (ƙ = 0.01) and high agreement (ƙ = 0.90), depending on familiarity with the scoring system and experience [30]. Although the two observers in our study were very experienced veterinarians, they had not used the scoring system before, which may explain the lower agreement compared to the trained observers in the original study. Also, the visual scoring system was modified, and several criteria present in the original scoring system by Main et al. [30] were removed to preserve true blinding of the video-assessors (Table 2). The items “initial response to human presence”, “pig’s response after opening gate” and “behaviour of individual within group” from the original protocol were not used, since this would require marking individuals to identify them within the group. Observers would then be able to recognize individuals on subsequent timepoints which may bias their scoring. The usability and sensitivity of visual scoring is enhanced by the addition of these behavioural criteria, although this was not confirmed in spontaneous OA in pigs [8]. Disuse of these criteria for scoring lameness may have resulted in the reduced performance of visual scoring. It is possible that particular aspects of lameness such as head bobbing are difficult to observe in pigs [30].

Another potential confounding factor, is the extensive training of the piglets to evenly trot for a click and food reward. Locomotion induced pain and a sweet treat reward at the destination, especially in a species as the pigs can pose two strongly opposing motivators to display outwardly visible pain behavioural. During the training, initially several animals were trained together, which may have induced fierce competition for the sweet treat reward, shaping getting to the sweet treat as a very strong motivator. Perhaps this point is best illustrated with a standard practice in neonatal care, where a sweet tasting liquid is administered orally, just prior to an invasive painful stimulus (like a heel prick) [42]. This on account of these (premature) neonates not showing, or showing to a lesser extent the characteristic pain face when being administered sucrose or a sweet tasting liquid. Similar to ambulation, a neonatal pain face originates from a highly shaped pattern of muscle contractions of which the coordination is mainly spinally mediated, with top down cerebral modulation [42]. This at least raises the possibility that outwardly displayed pain behaviour is susceptible to cerebral modulation induced by a “tug of war” between two strong yet opposing motivators. 

Scoring from video recordings could also be more difficult than expected, however, movie clips have been used to score lameness in several other studies [43,44,45,46]. Observers commented on some of the recordings as “difficult to score”, which may mean that some of the errors are due to having to score from movies instead from live observations. On the other hand, in veterinary practice conditions are usually much worse, with low lighting intensities and crowded stables. The identification of lame animals by visual inspection is likely even more difficult under on-farm conditions than under the optimised experimental conditions of the present study. Consequently, we expect that under on-farm conditions, many (sub)clinically lame animals probably remain undetected.

MIA injection did not result in a lasting clinical lameness when assessing it visually. This means that if OA is induced using MIA, most animals will not be lame enough to be selected as candidates for treatment by the farmer or veterinarian. 

When scoring lameness in pigs visually, the stoic nature of the species [8] likely contributed to the low predictability and correlation between the degree of joint damage induced and degree of lameness scored.

Eight animals from the MIA group needed rescue analgesia once due to clinical lameness scores of 4 or higher several hours after intra-articular injections were administered. It is possible that rescue analgesia masked more extreme differences on the first day after injection. The aim of this study, however, was to induce clinical lameness which would progress into a chronic condition. It is unlikely that a single clinical dose of buprenorphine would influence lameness scores for more than 12 h, let alone for 28 days afterwards. The pain behaviour suppressive effect of buprenorphine reported in experimental pain studies is dose dependent [47]. For the dose administered as rescue analgesia in this study the expected duration of effect is around 7 h. Although excessive dose escalation may inconsistently increase duration of effect considerably [48], in the clinical dose range the duration of effects does not exceed 12 h [49].

Another possible explanation for visual scoring not being able to detect severe lameness over a longer period is that the dose of MIA used in this study was not high enough to induce more severe lameness. Data from horses and dogs [50,51] indicated that MIA should not be dosed on body weight. Therefore, the MIA-dose we administered to our pigs (20 mg in an 80 mg/mL solution) was based on a pilot study using three doses: 10 mg, 20 mg or 40 mg (Appendix A). Both functional impairments and histological evidence of osteoarthritis development at day 11 were seen in the 20 and 40 mg dose, however, changes in the 40 mg dose were more severe and expected to progress to unacceptable levels of lameness and possibly ankyloses during the intended study period of 68 days post intra-articular injection. The 20 mg dose induced changes that were more comparable to previously reported changes in rats on day 10–14 [52] and was therefore selected for the study reported here.

##### Kinetic Data 

Our criterion for face validity in a pig model of MIA induced OA concerning kinetic characteristics was that pigs should show decreased weight-bearing in both the acute and chronic phase in the leg with experimentally induced OA. The kinetic data in our study showed a biphasic decrease in weight-bearing in the MIA injected group. This is consistent with the well-characterized development of MIA-induced functional osteoarthritic changes in rats [53]. Immediately after intra-articular MIA-injection in the knee, rats showed a marked decrease in weight-bearing that was most pronounced at 4 days post-injection. By 7 days post-injection, weight-bearing had almost entirely normalized. At day 14 post-injection, however, a second less pronounced stage of decreased weight-bearing developed which did not resolve for the remainder of the study (28 days post-injection). This biphasic decrease in weight-bearing was also observed in rats by Fernihough et al. [54]. 

In conclusion, our criterion for face validity was partly fulfilled. Weight-bearing measured with an objective method decreased, however visual inspection did not identify a significant increase in lameness after day 1. This means that although intra-articular MIA injection can be used to induce lameness, this lameness is not consistently severe enough to model animals that are identified as candidates for treatment under on-farm conditions.

### 4.2. Predictive Validity

Longitudinal kinetic gait assessment data of our model was consistent with the rat MIA-model, which is favoured as a model to assess disease modifying (oral) drug (DMOD) therapies in OA [35]. Although predictive validity of our pig model was not part of the present study, we recently performed a study using this animal model [55]. We investigated, using the present pig model, whether a disease modifying oral drug (DMOD) therapy is a feasible option for intervention. Despite caution for non-steroid anti-inflammatory drug (NSAID) treatment in conditions requiring bone remodeling is sometimes advocated [56], no adverse side-effects of NSAID (Meloxicam) treatment were detected in our pig model. Daily treatment for 48 consecutive days with oral Meloxicam (Metacam 15 mg/mL oral suspension for pigs, Boehringer Ingelheim Vetmedica, Rohrdorf, Germany), a potential osteoarthritis DMOD, in healthy control animals (intra-articular saline injections) did not affect gait, and further did not reveal adverse effects in growing pigs. Weight gain, gait, trabecular bone parameters, growth plate morphology, gastrointestinal integrity, and kidney histology were al shown to be unaffected by long-term Meloxicam treatment. These findings are not yet supportive evidence for the feasibility of daily DMOD therapy with Meloxicam in porcine osteoarthritis. However, when Meloxicam as DMOD therapy following intra-articular MIA injections would be assessed in our model, confounding pain or gait abnormalities due to aberrant skeletal development would not be expected. As part of the iterative process of model evaluation, other putative therapeutics should be tested in this model to further establish its predictive validity.

### 4.3. Construct Validity

For the MIA model to have construct validity, the mechanisms underlying the main clinical symptom lameness should be comparable to the naturally-occurring disease. 

We only performed morphological assessment of joint condition at necropsy on day 68 and therefore we do not have information regarding the development of structural joint changes prior to day 68. A non-invasive imaging technique such as MRI may have provided insight into the dynamics of morphological joint changes. This would have enabled comparison with the kinetic measurements, without losing animals for the later timepoints, as is the case with necropsy of some animals at an earlier timepoint. In a pilot study (Appendix A), a board-certified veterinary radiologist blinded for treatment assessed the carpal joint of pigs injected with MIA and saline solution 11 days previously with a 1.5 Tesla closed model MRI post-mortem just prior to histopathology. It proved difficult to relate the MRI images of the freshly obtained affected and control carpal joints with the histopathology performed the same day. Furthermore, anaesthesia required for imaging during a longitudinal in vivo experiment might have caused residual spinal anti-nociceptive effects [57,58]. This may interfere with subsequent clinical scoring of lameness, particularly in the earlier time-points, when interval times are short. We therefore decided not to include imaging techniques requiring anaesthesia in our final study design.

In the case of OA, lameness is initially due to (inflammatory) pain. In advanced cases, altered range of motion due to bony changes is an additional cause for impaired locomotion [8]. Although inflammation and joint damage are generally considered to contribute to joint pain, radiographic grading of the severity of osteoarthritis in humans often does not match the amount of pain reported [12,59].

Joint pain associated with OA in prone individuals is a progressive, fluctuating, mixed, and ultimately severely debilitating chronic pain complex [60,61,62,63]. Generally, initial acute inflammatory pain with straightforward relation between tissue damage and pain sensation becomes progressively unhinged due to functional and structural changes in the nociceptive pathways. Around the time that subchondral bone becomes exposed due to full thickness cartilage erosion, a neuropathic pain component [64] accompanies and interacts with the inflammatory and mechanical pain component [9]. A complex, mixed chronic progressive pain syndrome ensues. Supraspinal mechanisms meant to modulate ascending nociceptive input will ultimately fail and drive the affective-motivational and cognitive-evaluative components of the pain experience towards a debilitating condition often contributing to a general anhedonic and depressive state of mind [60]. As such a state is known to influence the perception of evoked (experimental) pain (such as is tested during human OA staging and OA pain studies) [59], it is hardly a surprise that poor correlation is consistently reported [59,65] between anatomical joint scores of affected joints and associated joint pain. Although decreased weightbearing assessed by kinetic analysis in the MIA group followed a bi-phasic pattern in agreement with the data in rat, at present we do not know if structural changes are the cause of this decrease.

Kinetic gait assessment revealed significant alteration in locomotion at timepoints representing the more acute as well as the chronic phase of MIA induced OA. However, with the visual scoring system used only the acute phase of pain-associated lameness was detected against the background of isotonic saline control treatment. Potentially, increasing the dose of intra-articular MIA could improve the performance of our model with regards to visual lameness scoring during the more chronic phase of MIA-induced OA. 

## 5. Conclusions

Although we feel that naturally-occurring animal models are the best reflection of the actual disease in both humans and animals, they have clear limitations [10]. Especially in larger animals and with OA, disease development, and progression, may take months to years. It may affect multiple sites in the body and during development uncontrolled comorbidities may ensue, negatively impacting a controlled study of an isolated site of disease manifestation. And first and foremost, in many animals the disease will never develop at all [35].

Although the MIA model evaluated in this study may be useful for some applications, it also has limitations. The aim was to develop a model the acute and chronic phase of OA to ultimately assess the effects of interventions on lameness in commercially housed pigs. Although objective parameters for weight-bearing showed that lameness was present in MIA-treated animals in both the acute and chronic phase of induced OA, visually many animals were considered “sound” by two experienced observers. Therefore, it is unlikely that the detected level of lameness by kinetic analysis in our model would ever be identified on-farm as in need of intervention. The decreased weight-bearing however, can be interpreted as a behavioral indicator of pain [8] and therefore the MIA model as reported here may have its merit as a general model for pain in pigs. 

## Figures and Tables

**Figure 1 animals-09-00405-f001:**
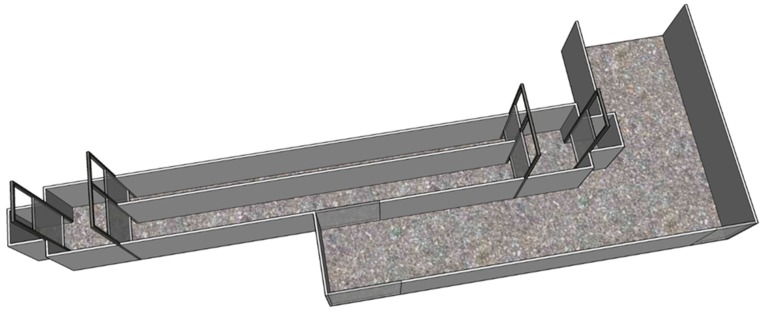
Schematic illustration of the experimental setup of the runway containing a pressure mat. The pressure mat was positioned exactly half-way in the runway at the back (furthest away from the penmates in the holding area). The cameras were positioned on the top of the frames of the guillotine doors on either side of the runway.

**Figure 2 animals-09-00405-f002:**
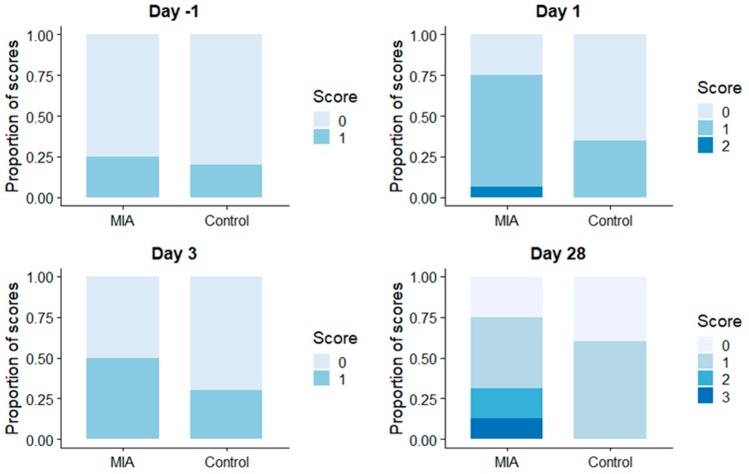
Proportion of different visual lameness scores modified from (see Table 2 for scored parameters) on specified timepoints during the experiment. The time-point designated panels represent blinded observer assigned offline lameness score distribution in the MIA treatment (MIA) and Saline Control treatment (Controls) groups. Day-1: on the day before intra-articular injection; Day 1, Day 3 and Day 28: one day, 3 days, and 28 days respectively after intra-articular injections.

**Table 1 animals-09-00405-t001:** Litter of origin, gender and treatment group of selected piglets.

ID	Litter	Gender	Group
1	1	F ^1^	MIA ^3^
2	1	M ^2^	MIA
3	2	F	MIA
4	2	F	MIA
5	3	M	MIA
6	3	F	MIA
7	4	F	MIA
8	4	F	MIA
9	5	F	MIA
10	5	F	MIA
11	1	M	SC ^4^
12	1	F	SC
13	2	M	SC
14	2	M	SC
15	3	F	SC
16	3	M	SC
17	4	F	SC
18	4	M	SC
19	5	M	SC
20	5	F	SC

^1^ F = female; ^2^ M = Male; ^3^ MIA = Monosodium Iodoacetate treatment; ^4^ SC = Saline Control treatment.

**Table 2 animals-09-00405-t002:** Experimental procedures.

Day Relative to Joint Injection	Age of Animals (days)	Performed Procedures
–21	28	Selection of litters and piglets; Weaning and transportation to experimenteal facility
–21 to –1	28–49	Habitation and training (to personnel and being alone in the different test set-ups)
–1	49	Baseline measurements: Video recording of gait (for offline visual scoring) Pressure mat measurements
0	50	General anaesthesia for intra-articular injection: MIA (10 animals, 2 ♂ ^1^, 8 ♀ ^2^)
		Isotonic saline (10 animals, 7 ♂, 3 ♀)
1	51	Day 1 measurements: Video recording of gait (for offline visual scoring) Pressure mat measurements
3	53	Day 3 measurements: Video recording of gait (for offline visual scoring) Pressure mat measurements
7	57	Day 7 measurements: Pressure mat measurements
14	64	Day 14 measurements: Pressure mat measurements
21	71	Day 21 measurements: Pressure mat measurements
28	78	Day 28 measurements: Video recording of gait (for offline visual scoring) Pressure mat measurements
56	106	Day 56 measurements: Pressure mat measurements
68	118	General anaesthesia for euthanasia; necropsy

^1^ ♂ being an intact male; ^2^ ♀ being an intact female.

**Table 3 animals-09-00405-t003:** Visual scoring protocol modified from Main et al. [30].

Lameness Score	Standing Posture	Gait
**0**	Pig stands squarely on 4 legs.	**Even strides**^1^. Caudal body sways slightly while walking. Pig is able to accelerate and change direction rapidly
**1**	As for score 0.	**Abnormal stride length** (not easily identified). **Movements no longer fluent** (pig appears stiff). Pig still able to accelerate and change direction.
**2**	**Uneven posture.**	Shortened stride. Lameness detected. Swagger of caudal body while walking. No hindrance in pig’s agility.
3	**Uneven posture. Will not bear weight on affected limb** (appears to be standing on toes).	Shortened stride. Minimum weight-bearing on affected limb. Swagger of caudal body while walking. Will still trot and gallop.
**4**	**Affected limb elevated off floor**. Pig appears visibly distressed.	**Pig may not place affected limb on the floor while moving.**
**5**	Will not stand unaided.	**Does not move.**

^1^**Bold type** identifies the defining criteria that must be present to assign a score. Normal type shows supporting criteria that are useful for assigning a score.

**Table 4 animals-09-00405-t004:** Mean values ± SD on each day for asymmetry indices (ASI) of Peak Vertical Force (PVF) and Vertical Impulse (VI) in the MIA (n = 8) and SC (n = 10) groups.

Day	Parameter	MIA	SC
–1	PVF	2.94 ± 20.59	−0.27 ± 18.88
	VI	7.36 ± 24.55	1.52 ± 20.30
1	PVF	−36.79 ± 22.94	−28.55 ± 25.42
	VI	−43.47 ± 22.98	−28.06 ± 27.59
3	PVF	−27.08 ± 25.41 **^1^**	-0.38 ± 16.77
	VI	−33.16 ± 19.28 **^1^**	−3.44 ± 19.10
7	PVF	−18.32 ± 30.12	−3.64 ± 25.02
	VI	−19.90 ± 30.58	−4.99 ± 22.05
14	PVF	32.84 ± 23.58 **^1^**	2.31 ± 17.82
	VI	−33.46 ± 26.89 **^1^**	4.21 ± 18.63
28	PVF	−32.84 ± 32.67 **^1^**	14.69 ± 12.03
	VI	−32.58 ± 34.30 **^1^**	16.84 ± 8.63
56	PVF	−44.14 ± 50.09 **^1^**	−3.10 ± 14.70
	VI	−40.03 ± 56.93 **^1^**	−3.83 ± 17.06

**^1^** Significant difference from SC group on the same day.

**Table 5 animals-09-00405-t005:** Estimates and standard errors for intercept and differences with reference category Peak Vertical Force (PVF) for each explanatory variable in the final model.

Explanatory Variable	Estimate	Standard Error	*p*-Value
(Intercept) ^1^	2.94	8.71	0.737
Day 1	−39.73	11.58	0.001
Day3	−30.01	11.58	0.011
Day7	−21.26	11.58	0.070
Day 14	−35.77	11.58	0.003
Day28	−35.78	11.58	0.003
Day56	−47.08	11.58	0.000
SC group	−3.21	11.69	0.787
Day 1 × SC group	11.45	15.54	0.463
Day 3 × SC group	29.90	15.54	0.057
Day 7 × SC group	17.89	15.54	0.253
Day14 × SC group	38.35	15.54	0.015
Day 28 × SC group	50.75	15.54	0.002
Day56 × SC group	44.25	15.54	0.005

^1^ Mean PVF for reference category (Day-1, MIA group); SC = saline control

**Table 6 animals-09-00405-t006:** Estimates and standard errors for intercept and differences with reference category Vertical Impulse (VI) for each explanatory variable in the final model.

Explanatory Variable	Estimate	Standard Error	*p*-Value
(Intercept) ^1^	8.382.94	8.50	0.326
Day 1	−50.83	11.97	0.000
Day3	−40.52	11.98	0.001
Day7	−27.26	12.02	0.026
Day 14	−40.83	12.16	0.001
Day28	−39.94	12.66	0.002
Day56	−47.40	14.46	0.002
SC group	−1.96	11.69	0.869
Day 1 × SC group	21.24	16.06	0.189
Day 3 × SC group	35.54	16.08	0.029
Day 7 × SC group	20.74	16.13	0.202
Day14 × SC group	43.51	16.31	0.009
Day 28 × SC group	55.25	16.99	0.002
Day56 × SC group	42.04	19.41	0.033

^1^ Mean VI for reference category (Day-1, MIA group); SC = saline control

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
