# Peer review of "A Monosodium Iodoacetate Osteoarthritis Lameness Model in Growing Pigs"

_animals, 2019, doi:10.3390/ani9070405_

Round 1
Reviewer 1 Report
General comment :The manuscript “ A monosodium iodoacetate osteoarthritis lameness model in growing pigs: assessing face and construct validity with respect to lameness” could be interesting, because the contents addressed in this study are worthy of investigations respect to their scientific and practical application. The manuscript deals with the validation of an osteoarthritis lameness model in growing pigs in order to study the naturally-occurring osteoarthritis and its progression by understanding the functional and structural aspects of the disease. Nevertheless, the face validity of the model for osteoarthritis under field conditions was not entirely met and the conclusions only suggest the usefulness of the model as a general model for pain in pigs. Probably, the lack of useful results responding to the proposed aims of the study is dependent on the weakness of the study design. The manuscripts is in fact crammed with a large introduction on the description of how to define an “animal model” and its validity, while the study design, the methods adopted and mainly the data presented result incomplete respect to this definition and to the proposed construct validity of the model as presented in the aim of the study.
The English used in the paper must be reviewed by a native speaker. In the same time some minor mistakes could be reviewed by the Authors.
The current manuscript is acceptable for publication after minor revision.
Title: The title is not appropriate. It could better rephrased in relation to the effective resukts obtained and conclusions.
Simple Summary and Abstract: The Simple Summary and the Abstract does not effectively represent the data presented in the study. The Abstract does not entirely correspond to Simple Summary.
Introduction: This section is too long. It includes an excessive discussion on the definition of “animal model” and on the quantification of lameness in pigs, while no presentation of the available actual models to measure pain in pigs is discussed. It should be better to concisely introduce the proposed aims of the study. Moreover, the introduction does not correspond to the data presented, while the results are mainly behavioural data and histopathological data are not presented.
Material and methods: The experimental design is not well constructed. The treatment and control groups of animals have not a well-balanced composition, concerning gender. The criteria adopted in order to choose the dose of MIA injected are not discussed. Past-published results of the pharmacological treatment of the animal model are only briefly mentioned. Necropsy and histopathology methodology are presented, although the results of the structural and histopathological effects are not extensively presented. The statistical analysis is adequate.
Results: Although there is an interesting amount of data and the corresponding tables are clear, the results presented are incomplete respect to the aim of the paper, the methodology described and the conclusions reported. The results of the structural and histopathological effects of monosodium iodoacetate injected are not extensively presented and discussed. Only subjective visual scoring and gait kinematics analysis were largely considered.
Discussion: The comments and mainly the conclusions reported in discussion are not completely pertinent to the aim of the paper and results achieved. Concerning the conclusions proposed, a more exhaustive discussion on pain behaviour of pigs and its evaluation and on the relationships of pain behaviour and pathology described should be requested.
References: They are up-to-date, complete and appropriate, although references on pain indexes usually adopted on pigs should be lso presented.
Reviewer 2 Report
Some minor changes should be made:
27: use carpal rather than wrist
177: indicate briefly how healthy, sound piglets were selected
191: Handling and habituation as the degree of exercise is likely to have an effect on outcomes this should be indicated in a bit more detail
224: justify why the left carpal joint was selected rather than random sides. There may have been some biases in visualising the gait
230: Was there any sign of joint fluid when the needle was inserted?
252: as the protocol has varied from Main et al. this should be labelled as such and more detail (in the Discussion) what has been modified.
340: use the term stifle or femorotibial rather than "knee"
341: Are the 2 without OS changes the same as the 2 without histopathological changes (353) What did they score in the visual assessments?
370/372/375/387: mentions the "OS group". I assume that is the MIA group? It should be consistent.
391: 14d VI MIA has an asterisk - it should be indicated why.
